# Inflammatory Depression—Mechanisms and Non-Pharmacological Interventions

**DOI:** 10.3390/ijms22041640

**Published:** 2021-02-06

**Authors:** Klara Suneson, Jesper Lindahl, Simon Chamli Hårsmar, Gustav Söderberg, Daniel Lindqvist

**Affiliations:** 1Department of Clinical Sciences Lund, Psychiatry, Faculty of Medicine, Lund University, 221 85 Lund, Sweden; jesper.lindahl@med.lu.se (J.L.); gustav.soderberg@med.lu.se (G.S.); daniel.lindqvist@med.lu.se (D.L.); 2Office for Psychiatry and Habilitation, Psychiatric Clinic Helsingborg, Region Skåne, 252 23 Helsingborg, Sweden; 3Office for Psychiatry and Habilitation, Psychiatric Clinic Lund, Region Skåne, 221 85 Lund, Sweden; Simon.ChamliHarsmar@skane.se; 4Office for Psychiatry and Habilitation, Psychiatry Research Skåne, Region Skåne, 221 85 Lund, Sweden

**Keywords:** depression, inflammation, dysbiosis, n-3 PUFAs, exercise

## Abstract

Treatment of depression is hampered by the failure to identify distinct symptom profiles with distinct pathophysiologies that differentially respond to distinct treatments. We posit that inflammatory depression is a meaningful depression subtype associated with specific symptoms and biological abnormalities. We review several upstream, potentially causative, mechanisms driving low-grade inflammation in this subtype of depression. We also discuss downstream mechanisms mediating the link between inflammation and symptoms of depression, including alterations in dopaminergic neurotransmission and tryptophan metabolism. Finally, we review evidence for several non-pharmacological interventions for inflammatory depression, including probiotics, omega-3 fatty acids, and physical exercise interventions. While some evidence suggests that these interventions may be efficacious in inflammatory depression, future clinical trials should consider enriching patient populations for inflammatory markers, or stratify patients by inflammatory status, to confirm or refute this hypothesis.

## 1. Introduction

Several lines of preclinical and clinical evidence connect systemic low-grade inflammation to major depressive disorder (MDD). A large number of studies have now shown that depressed patients have increased mean levels of blood and cerebrospinal fluid (CSF) inflammatory markers, including high-sensitivity C-reactive protein (hs-CRP) [1,2,3]. Importantly, elevated blood levels of hs-CRP are associated with relevant brain biomarkers of MDD including decreased corticostriatal connectivity [4], increased basal ganglia glutamate [5], and elevated CRP in CSF [6], confirming the usefulness of peripheral blood hs-CRP as an accessible and standardized biomarker for inflammatory depression. There is also evidence for inflammation being a causal factor in a subgroup of depression and not just an epiphenomenon, as illustrated by the potential of interferon-alpha (INF-α) treatment to induce depression in susceptible individuals [7]. Despite solid scientific support for the “inflammation hypothesis of depression”, at least in certain subgroups of patients, several unresolved issues remain before this can lead to clinically meaningful outcomes. Firstly, inflammation is certainly not specific for MDD, and not all MDD patients have signs of inflammation. In recent years, it has become increasingly clear that MDD is not an inflammatory disorder per se, but instead there is evidence for a subset of MDD patients with increased inflammation. This subtype of so-called inflammatory depression, which has been poorly studied to date, has been associated with treatment resistance to selective serotonin reuptake inhibitors (SSRIs), and with obesity, medical co-morbidities, and a specific symptom profile characterized by sleep and appetite disturbances, fatigue, and lack of motivation [8,9]. Another important question that has not been fully addressed is what causes the low-grade inflammation observed in this subgroup of MDD. This knowledge is needed in order to identify preventive measures and treatment targets for inflammatory depression.

Inflammatory depression might be meaningful to distinguish from other, non-inflammatory, subtypes of depression in order to personalize antidepressant treatment. Here, we review several upstream and downstream mechanisms involved in the pathophysiology of inflammatory depression, and potential interventions to treat this condition. Since previous reviews have mostly covered either the anti-inflammatory effects of conventional antidepressants [10], or antidepressant efficacy of various anti-inflammatory medications [11,12], our focus will be on nutraceuticals and lifestyle interventions, and their potential role in treating inflammatory depression.

## 2. Mechanisms of “Inflammatory Depression”

The causes of the chronic low-grade inflammation observed in some cases of MDD are not fully understood, and are likely to vary from one individual to another. Candidate mechanisms that have been explored in recent years include, but are not limited to, autoimmune disorders, infections, physical inactivity, poor diet, and genetic predisposition [8,13]. Cytokine signaling triggers a multitude of downstream biological effects on neuroendocrine, monoaminergic, and oxidative stress systems to name a few [8]. Alterations in these systems may subsequently lead to “sickness behavior” (depressed mood, fatigue, loss of appetite), which in some cases progresses to manifest clinical depression [13,14]. Such depression-related behavioral changes could in turn reinforce negative health behaviors including stress, physical inactivity, and poor dietary habits, establishing a vicious cycle generating more inflammation and a progressive worsening of depressive symptoms, as illustrated in Figure 1.

In the following sections, we describe several upstream, potentially causative, mechanisms of inflammatory depression including alterations in the gut microbiome (dysbiosis) and an imbalance in the omega-6/omega-3 ratio. We also describe selected downstream mechanisms, potential mediators between inflammation and depressive symptoms, focusing on inflammation effects on tryptophan and dopamine metabolism and neurotransmission.

### 2.1. Upstream Mechanisms; from Lifestyle to Inflammation

#### 2.1.1. Dysbiosis

Dysbiosis refers to an imbalance in the gut microbiota, including a reduction in microbiota diversity and fewer beneficial microbes, as often seen in somatic conditions such as obesity and type 2 diabetes [15]. Several lifestyle factors including dietary habits [16], exercise [17,18], and smoking habits may influence microbiota composition [19]. Dysbiosis is central in the theory that alterations in the gut-brain axis may influence the onset and maintenance of inflammatory depression [20]. One important mechanism in this regard is that dysbiosis may promote increased intestinal permeability (referred to as “leaky-gut” and illustrated in Figure 2), leading to bacterial translocation across the intestinal barrier into the circulation [21]. This, in turn, leads to a Toll-like receptor 4 (TLR4)-dependent immune response to lipopolysaccharides (LPS). LPS are usually found in the outer membrane of Gram-negative bacteria, and when they pass from the gut lumen into the circulation, they bind to LPS-binding protein (LBP), and this complex in turn binds to immune cells triggering secretion of cytokines [21,22]. Elevated levels of LPS have been detected after intake of high-energy meals [21]. Moreover, one study found that MDD subjects have increased serum anti-LPS antibodies compared to controls [23], further supporting the role of increased gut permeability and bacterial translocation across the intestinal barrier, resulting in systemic inflammation, in MDD pathophysiology.

There is also evidence that certain gut microbial products may counteract this LPS-induced immune response. For instance, short-chain fatty acids (SCFAs) are neuroactive bacterial metabolites with the ability to translocate from the gut lumen to the systemic circulation and subsequently cross the blood-brain barrier (BBB) [21]. SCFAs are considered to have anti-inflammatory properties during certain conditions, mitigating LPS-induced immune responses [24]. Moreover, SCFAs may have beneficial effects on the integrity of the intestinal barrier as well as the BBB [21,25], suggesting that they have an important intermediary role in how dysbiosis may alter neurobiological functions.

To the extent that changes in the gut microbiota may cause depressive symptoms in certain patients, it is crucial to determine which factors mediate this relationship. Neuroinflammation has been implicated in MDD pathophysiology [26], and is partially mediated by microglia activation. Animal studies suggest that gut microbiota regulate microglial development and function [27]. Erny et al. showed that germ-free mice display global defects in microglia. They also reported that eradication of microbiota significantly changes microglia properties including compromised maturation and cell shape [27]. Recolonization with a more complex microbiota was able to restore microglia features in these mice [27]. Interestingly, SCFA administration was also able to restore some aspects of microglia immaturity and malformation [27]. Thion et al. recently corroborated the impact of the gut microbiome on microglial function when they showed that maternal microbiome characteristics influence microglial properties during prenatal stages in mice, and that microbiome depletion has effects on microglial properties [28]. The exact mechanisms by which the gut microbiome may influence microglia are not clear but may involve a dysbiosis-induced leaky-gut, and subsequent translocation of microbiome and inflammatory response [21]. There is also animal data suggesting that the gut microbiota influence tryptophan and kynurenine pathway metabolism [29,30]. For instance, certain gut bacteria, such as *Lactobacillus*, secrete indoles [31]—tryptophan metabolites that have been shown to augment gut barrier integrity and have anti-inflammatory properties [32], in part by counteracting the effects of LPS [33]. Taken together, there is evidence, mainly from preclinical studies, suggesting that the gut microbiota can affect various physiological and neurological processes, relevant for depression pathophysiology, via both direct and indirect pathways. Further research on human subjects is needed before this knowledge can be translated to a clinical context.

#### 2.1.2. Increased Dietary Omega-6/Omega-3 Ratio

Polyunsaturated fatty acids (PUFAs) omega-6 (n-6) and omega-3 (n-3) have a multitude of immune-modulating effects, both conjointly with [34] and separately [35,36] from those related to dysbiosis. As exemplified below, n-3 and n-6 effects are predominantly described as anti- and pro-inflammatory, respectively [35]. In Western societies, dietary intake of n-6 sharply exceeds that of n-3, which may promote a state of systemic low-grade inflammation [37]. This is reflected in increased intake of, e.g., beef, chicken, and egg, which are animal-based sources of preformed n-6, and a decreased intake of fish (source of preformed n-3). There are also plant-derived sources of n-3 and n-6, which are considered less efficient as they only provide precursors requiring several intermediate metabolic steps before n-3 and n-6 synthesis [38].

As reviewed by Calder [39], N-3 PUFAs, eicosapentaenoic acid (EPA) and docosahexaenoic acid (DHA), have several important anti-inflammatory properties, including (i) decreased chemotaxis of neutrophils and monocytes, (ii) decreased expression of adhesion molecules on the surface of immune cells and in the circulation, (iii) decreased production of prostaglandins, and (iv) inhibition of T-cell proliferation. EPA and DHA are naturally embedded in phospholipid membranes and are both metabolized via enzymes including cyclooxygenase (COX) and lipoxygenase (LOX), generating anti-inflammatory products such as the EPA-derived E-resolvins and DHA-derived D-resolvins (both via COX-2) [38,39]. A detailed review of the anti-inflammatory effects of resolvins and associated EPA/DHA-products are beyond the scope of this review, but have been described elsewhere [40].

Enzymes COX and LOX are, in addition to being involved in the anti-inflammatory metabolic pathways associated with EPA and DHA described above, also responsible for breakdown of other PUFAs resulting in pro-inflammatory metabolites. For instance, arachidonic acid (AA) is a n-6 PUFA exerting diverse immunomodulating effects, mainly of pro-inflammatory nature [35]. When released from immune cell membranes, free AA is a major substrate in the synthesis of eicosanoids, including prostaglandins (PG), thromboxanes, leukotrienes, and lipoxins [38]. Eicosanoids are part of an intricate, not yet fully understood system that influences the inflammatory response in disparate directions. Eicosanoid immune modulation depends on various factors such as the timing of interactions between specific eicosanoids and immune cells and what type of receptors those immune cells express [35]. Hence, prostaglandins may exert pro-inflammatory effects in one setting and anti-inflammatory effects in another, as reviewed by Tilley et al. [41]. Therefore, AA-degradation (commonly and hereafter referred to as the AA cascade) and the function of formed eicosanoids is of importance for the duration and intensity of immune responses [35]. Briefly, the AA cascade is initiated through mobilization of AA from membranes by phospholipase A2 (PLA2), which is activated by cytokines such as tumor necrosis factor alpha (TNF-α) [42]. In subsequent steps of the cascade, free AA will in part be converted, by, e.g., COX-1, -2, LOX, or cytochrome P450, to eicosanoids [43]. Interestingly, some of these enzymes are also involved in the metabolism of n-3 PUFAs generating anti-inflammatory mediators. On the contrary, degradation of AA by COX and LOX may render pro-inflammatory eicosanoids, hence n-3 and n-6 PUFAs may compete for available enzymes resulting in either an anti-inflammatory or pro-inflammatory net outcome [36], as illustrated in Figure 3. An example of the latter is production of the prostaglandin PGE2, derived via COX, which enhances production of pro-inflammatory cytokine interleukin-6 (IL-6) in immune cells [35,38]. Moreover, the metabolization of AA by COX or LOX may generate reactive oxygen species (ROS) further reinforcing the inflammatory response [44]. Taken together, the ratio of dietary n-6 to n-3 influence the balance between pro- and anti-inflammatory effects and may thus mediate the effects between diet, inflammation, and depressive symptoms. As described further below, these pathways have therefore been suggested as potential treatment targets in inflammatory depression.

To our knowledge, a potential connection between the AA cascade, depression, and inflammation has so far been sparsely explored. An exception is a study by Su et al. investigating the association between *PLA2*-gene polymorphisms and risk of IFN-α induced depression among 132 patients with chronic hepatitis C infection undergoing treatment with IFN-α [45]. Twenty-eight percent of the subjects developed IFN-α induced depression during a 24-week follow-up period. Subsequent analysis of *PLA2* polymorphisms identified a genotype associated with a significantly elevated risk for developing depression. Intriguingly, subjects carrying the “risk” *PLA2*-genotype had significantly lower EPA levels in erythrocyte membranes compared to subjects with other genotypes. This may indicate that this *PLA2*-genotype is associated with an increased risk for developing inflammatory depression, and that this is mediated via n-3 levels, as detected in cell membranes. While mechanistically interesting, this study lacked measurements of n-6 PUFAs in membranes, which would have been of interest in relation to PLA2, and these results need to be replicated in independent samples. There are also reports of elevated levels of PLA2 in patients with various psychiatric illnesses [46], and a potential link between the AA cascade, bipolar disorder, and the effects of mood-stabilizers has been repeatedly investigated [43,47]. Interestingly, lithium treatment in mice leads to a suppression of AA turnover, potentially through decreased expression of brain PLA2 [48]. In sum, the effects of an increased n-6/n-3 ratio may be of relevance for the pathophysiology of inflammatory depression. As reviewed in Section 3.1, several studies have investigated the antidepressant effects of n-3 fatty acid supplements, with various degrees of success.

### 2.2. Downstream Mechanisms; from Inflammation to Symptoms

#### 2.2.1. The Kynurenine Pathway of Tryptophan Metabolism

Upstream mechanisms (e.g., dysbiosis and increased n-6/n-3-ratio) generate a pro-inflammatory state from where further signal transduction, particularly cytokine-mediated, can affect various aspects of neural circuits central to behavior and mood [13,49]. For example, pro-inflammatory cytokines can modify serotonergic, glutamatergic, and dopaminergic circuits via changes in the metabolism of tryptophan. As illustrated in Figure 4, tryptophan is the precursor of serotonin, yet it may also be metabolized along what is called the kynurenine pathway (KP). Here we describe the KP and review some of the current knowledge of its involvement in the pathophysiology of inflammatory depression [50].

The kynurenine pathway (KP) involves a series of enzymatic steps in the metabolism of essential amino acid tryptophan and has been extensively studied in relation to both inflammation and depression [51,52,53]. In particular, focus has been placed on how pro-inflammatory cytokines (e.g., TNF-α) may activate the enzyme indoleamine 2,3-dioxygenase (IDO), which converts tryptophan to kynurenine in the first and rate-limiting step of the KP [54,55]. Metabolism along the KP occurs concurrently with serotonin synthesis. Thus, increased IDO activity, e.g., during a pro-inflammatory state, may reduce the conversion of tryptophan to serotonin [56], which in turn may be relevant for MDD pathophysiology. The downstream metabolites generated in the KP differ depending on cell type. KP activation in astrocytes generates kynurenic acid (kyn-a) whereas KP in microglia may generate either 3-hydroxykunurenine (3-HK) and quinolinic acid (quin-a), or anthranilic acid [55]. Quin-a is a neuroactive metabolite, with N-methyl D-aspartate (NMDA)-receptor agonistic properties [51,57]. Interestingly, the novel antidepressant Ketamine exerts effect on the glutamatergic NMDA-receptor, however antagonistically and in contrast to the effects of quin-a. Quin-a has several neurotoxic properties including the ability to generate reactive oxygen species (ROS) and enhance CNS-inflammation by increasing levels of chemotactic molecules [55,58]. These effects of KP activation, mediated via the quin-a generating “neurotoxic branch”, result in altered serotonin availability, alterations in glutamatergic neurotransmission, neurotoxicity, and oxidative stress and may all contribute to development of depressive symptoms [59]. Indeed, a clinical study on patients with suicidal depression reported elevated CSF quin-a levels in cases versus controls and a direct correlation between CSF IL-6 and CSF quin-a [59]. Consistent with a causative role of KP activation in inflammatory depression, O’Connor et al. showed, in a mice model of depression, that activation of IDO occurs in parallel to the development of depressive-like behavior after peripheral LPS-injections [60]. Further supporting a causal link between KP activation and depression, pre-LPS inhibition of IDO prevented depressive-like symptoms in the same animal model [60]. Furthermore, Smith et al. showed that a polymorphism of the IDO-encoding gene was associated with an increased risk of developing depressive symptoms during treatment with IFN-α for hepatitis C [61]. Whereas KP activation in microglia generates metabolites associated with the “neurotoxic branch”, KP activation in astrocytes generates metabolites with potentially neuroprotective properties such as kyn-a, see Figure 4. In opposite to quin-a, kyn-a is a NMDA-receptor antagonist and has also other neuroactive and potentially neuroprotective effects, including an antagonistic interaction with the alpha-7 nicotinic acetylcholine receptor (a7nAchR) [55,62]. Antagonism on the a7nAchR may attenuate both neuroinflammation and depressive-like behavior as shown in mice by Alzarea and Rahman [63]. Currently, the diverse effects of metabolism along different KP-branches and the relevance for clinical depression of their respective end products are not fully understood. Clinical studies on depressed patients have rendered inconsistent results, partly due to small sample sizes, and between-study differences with regards to symptom severity, medication status, sampling medium (blood, CSF, or urine) [55,64], and sometimes failure to take into account potential confounds such as diet and smoking habits [55].

To the best of our knowldege, only one study so far has set out to identify subgroups of depressed patients with an “immune-activated KP”, defined by higher levels of TNF-α and kynurenine/tryptophan-ratio (kyn/trp-r) as proxy markers for higher IDO-activity [65]. This study used factor analyses to identify a MDD subgroup with high TNF-α and kyn/trp-r, and these patients were more severely depressed and anhedonic, and they were less likely to respond to antidepressant treatment. Larger studies with similar innovative designs as this one are needed in order to further validate these results, to investigate inflammatory biomarkers in subgroups of depressed patients, and to relate biological signatures to clinical characteristics and treatment outcomes.

#### 2.2.2. Dopaminergic Neurotransmission 

Altered dopaminergic metabolism and neurotransmission has been suggested as a key mediator between systemic low-grade inflammation and certain depressive symptoms [66]. Of note, both dopaminergic and immune-related alterations map onto a certain clinical subtype of depression characterized by symptoms such as anhedonia, amotivation, fatigue, and psychomotor retardation [8,9,66]. Accumulating evidence suggests that the behavioral sequelae of inflammation, including motivational anhedonia and alterations in reward-seeking behavior (c.f. “sickness behavior”), are a direct consequence of the impact of cytokines on mesolimbic dopamine signaling [66]. For instance, fMRI-studies of depressed patients report a correlation between peripheral, low-grade inflammation (assessed by hs-CRP) and anhedonia mediated via decreased resting-state functional connectivity in the reward circuit [4]. Moreover, various types of immune challenges (to healthy volunteers, patients with hepatitis or animals) triggers motivational anhedonia [67], a blunted response to reward anticipation in the ventral striatum [68], and decreased striatal dopamine release and availability [8,69].

The molecular mechanisms by which inflammation may lead to a hypodopaminergic state are not fully understood. Suggested pathways involve inflammation-induced alterations in availability and function of enzyme co-factor tetrahydrobiopterin (BH4) [67]. BH4 is crucial in the conversion of tyrosine to L-DOPA which is the rate-limiting step in dopamine synthesis [67]. Increased inflammation contributes to oxidation of BH4, leaving decreased enzyme availability for dopamine synthesis, see Figure 5. In addition, BH4 is required in the formation of the ROS nitric oxide (NO), by inducible nitric oxide synthase (NOS). Expression of NOS is enhanced by inflammation, promoting BH4 utilization in the synthesis of NO at the expense of dopamine synthesis. Conjointly, these alterations of decreased dopamine synthesis and increased NO-production may reinforce inflammation-oxidative stress interactions relevant for the pathophysiology of depression [70]. In line with this, Kitagami et al. demonstrated that peripheral IFN-α injections decreased levels of BH4 and dopamine in the amygdala and raphe of rodents [71]. Interestingly, in the same study, BH4 levels were restored with the addition of a NOS-inhibitor, suggesting a decreased dopamine synthesis due to increased BH4 consumption related to to NO synthesis. Another study on patients undergoing treatment with IFN-α [72] showed that lower CSF levels of BH4 correlated with higher CSF IL-6. Moreover, plasma phenylalanine/tyrosine ratio, as an indirect proxy measure of BH4 activity, correlated negatively with CSF dopamine and positively with symptoms of fatigue. These results are in line with a pro-inflammatory and hypodopaminergic state being associated with a depression symptom profile of fatigue, motivational anhedonia, and similar, associated symptoms.

The biological network that includes dopaminergic circuits, inflammation, and oxidative stress also involves further, multidirectional, mechanisms. For instance, the metabolism of dopamine may render ROS as byproducts and thereby add to oxidative stress [73]. Oxidative stress may in turn enhance cytokine production through feedback mechanisms and by that, uphold an inflammatory state [74]. Moreover, the release of dopamine from presynaptic neurons is reduced during states of inflammation as pro-inflammatory cytokines decrease the expression of the vesicular monoamine transporter 2 (VMAT2) [67]. Under normal conditions, the action potential generates integration of dopamine-vesicles into the presynaptic membrane, which results in release of dopamine to synaptic cleft. VMAT2 enables vesicular storage of dopamine in synapses before releasing it into the synaptic cleft, thus less dopamine is released if VMAT2 expression is lower [73]. In addition to altering dopamine synthesis and release, immune-activation may also change expression and function of the dopamine transporter (DAT), e.g., through intra-cellular signaling pathways involving mitogen-activated protein kinases (MAPKs) [67]. MAPKs are activated by binding of pro-inflammatory cytokines IL-1B and TNF-α or LPS to their respective membrane receptors [75]. Further downstream actions of MAPKs are primarily mediated by activation of the transcription factor-family nuclear factor kB (NF-kB), resulting in pro-inflammatory actions such as changes in leukocyte activity and cellular metabolism [75]. Immune-activated cascades of MAPK may increase both DAT reuptake and expression, leading to decreased amounts of dopamine available in the synaptic cleft [67]. Such regulation of DAT through MAPKs was demonstrated by Moron et al., as inhibition of MAPKs decreased both transport capacity and cell surface expression of DAT, in synaptosomes of rat striatum and human embryonic kidney cells respectively [76].

In summary, a balance in dopaminergic circuits is of crucial importance to regulate several neurobehavioral functions such as hedonic response and psychomotor function. So far, most studies have investigated how dopaminergic signaling within the mesolimbic pathway mediates motivational anhedonia and related symptoms. Less focus has been paid to peripheral, non-CNS, inputs to the dopaminergic reward circuitries that could potentially influence dopaminergic neuron responsivity and thereby generate depressive symptoms. Accumulating evidence suggests that systemic low-grade inflammation is an important upstream mechanism linking a hypodopaminergic state to “anhedonic depression”, and future studies testing interventions targeting either inflammation or dopaminergic mechanisms should consider actively enrolling patients with this particular symptom profile to potentially advance precision psychiatry.

## 3. Therapeutic Implications

### 3.1. Omega-3 Fatty Acids as a Potential Treatment of Inflammatory Depression

Several clinical studies have shown that MDD patients, on a group level, have lower blood levels of n-3 PUFAs [77], suggesting that n-3 PUFAs may be involved in MDD pathophysiology and that supplementation of n-3 PUFAs may have an antidepressant effect in certain patients. Most randomized controlled trials (RCTs) investigating the antidepressant efficacy of n-3 PUFAs have, however, demonstrated only small-to-medium effects [78,79,80]. Although concrete clinical guidelines for the use of n-3 PUFAs in depression have been proposed [81], these nutritional supplements have not yet had a clinical breakthrough. In a recent meta-analysis [82], including a relatively homogenous group of patients with Diagnostic and Statistical Manual of Mental Disorders (DSM)-verified MDD, a beneficial antidepressant effect of n-3 PUFAs was found (effect size = 0.4). However, the authors noted that there was a large variation in the quality of the included studies and a potential risk of publication bias [82]. Interestingly, they reported that several study design-related factors, such as add-on design and higher dose of EPA in the intervention arm were associated with a greater likelihood of a successful antidepressant response.

A possible explanation for the inconsistent effect sizes across studies is that not all patients with MDD, but only a subset, benefit from n-3 treatment. Given the anti-inflammatory properties of n-3 PUFAs described above, it is possible that only those MDD patients with signs of low-grade inflammation may respond to this particular intervention. This would be consistent with one animal study showing that n-3 PUFA treatment prevents LPS-induced depressive-like behavior in mice, by suppressing neuroinflammation [83]. Further supporting a role for n-3 PUFAs in inflammatory depression, another study found that pre-treatment with n-3 PUFA has a protective effect on IFN-α induced depression [84]. In this RCT, 162 patients with hepatitis C were randomized to two weeks of pre-treatment with EPA, DHA, or placebo before starting IFN-α therapy. While both EPA and DHA delayed the onset of IFN-α induced depression, only EPA significantly reduced the incidence rate compared to placebo (10% vs. 30%), suggesting that EPA, more so than DHA, might be efficacious in treating inflammatory depression. Consistent with a specific role for EPA in the treatment of inflammatory depression, a recent RCT showed that EPA, but not DHA, had a large antidepressant effect in depressed subjects with signs of low-grade inflammation [85]. Interestingly, such beneficial effect was not seen in those MDD patients with low levels of blood inflammatory markers. Moreover, this study provided evidence of a dose-response effect with greater separation between EPA and placebo in subgroups of patients with more than one elevated inflammatory marker. For instance, patients with 4–5 elevated inflammation biomarkers, had remission rates (Hamilton Depression Rating Scale-17 score ≤ 7) of 40% for EPA and 25% for placebo, while patients without any elevated inflammatory marker had remission rates of 19% for EPA and 44% for placebo. These findings strengthen the hypothesis that, while n-3 PUFAs may not be a robust antidepressant treatment overall, it might be clinically useful in the subgroup of inflammatory depression. To confirm or refute these novel, and potentially clinically meaningful, findings, future well-designed clinical trials should either enrich patient populations for inflammatory markers or use “match/mismatch designs” with subjects stratified by inflammatory markers at baseline and then randomized to receive an anti-inflammatory intervention or not [8,86].

### 3.2. Dysbiosis, Inflammation, and Depression—Are Probiotics Efficacious in Inflammatory Depression?

Accumulating preclinical data suggest that dysbiosis and increased gut permeability are associated with depressive-like behavior in animals [21,29], including some evidence for a causal mechanism. Kelly et al. [30] showed that germ-free mice receiving a fecal transplant from depressed patients but not from non-depressed controls, develop depressive-like behavior [30]. Rats receiving a fecal transplant from a depressed human donor also showed a trend towards higher plasma CRP than those receiving fecal transplants from controls, suggesting low-grade inflammatory responses to “depressed” human microbiota compared to “healthy” human microbiota [30]. Furthermore, several animal models of stress reported reductions in richness and diversity of the gut microbiota and increased immune activation [87,88].

In addition to animal studies showing a link between dysbiosis, inflammation, and depressive symptoms, there are also several lines of clinical evidence supporting this. Reduced microbiota diversity is considered an indicator of dysbiosis and has been associated with obesity as well as cardiometabolic and autoimmune disorders [89]. The specific mechanisms linking dysbiosis to certain psychiatric and somatic disorders are not fully understood but may involve an induction of chronic low-grade inflammation [90], as further described below and illustrated in Figure 2 above. The normal human gastrointestinal tract has an abundance of the bacterial phyla Bacteroidetes and Firmicutes, representing approximately 90% of the bacteria present [91]. A preliminary study of MDD subjects reported reduced numbers of Firmicutes in MDD compared to controls [92]. Another study reported that *Lactobacillus*, a genus of Gram-positive bacteria belonging to phylum Firmicutes, was reduced in MDD relative to healthy controls [93]. Though interesting, these studies had several limitations including lack of dietary habit data and did not account for potential confounders including concurrent medications. A recent large-scale population-based study found reductions in genus *Dialister* and *Coprococcus* in individuals with suspected depression, and antidepressant medication moderated these effects [94]. This study also reported significant differences in *Lactobacillus* in individuals with suspected depression, namely an increase in these bacteria.

Already in 2005, Logan and Katzman [95] suggested that probiotics may potentially be used as novel adjuvant therapies for MDD, based on limited evidence from various conditions frequently comorbid with depression (e.g., irritable bowel disease, chronic fatigue syndrome, fibromyalgia). The antidepressant effects of probiotics have subsequently been tested in a number of studies, but results have been inconsistent which might depend on sample heterogeneity across studies, various rating scales and bacterial strains being used, and the fact that most studies have included healthy individuals rather than clinically depressed patients [96]. To the extent probiotics have a meaningful effect on mood, this seems to be larger in those with at least moderate severity of depressive symptoms at baseline, and smaller in healthy, non-depressed, individuals. Based on preclinical evidence of an anti-inflammatory effect and the ability to reduce intestinal permeability and protect against a “leaky gut” [97,98,99], *Lactobacillus* species have been widely used as probiotics. One study reported that the quantity of Lactobacilli is reduced in patients with MDD compared to healthy controls [93]. In mice, lower amounts of Lactobacilli have been associated with increased oxidative stress [100], another frequent finding in MDD [2]. Animal studies testing *Lactobacillus* probiotics reported positive effects on depressive and anxious behaviors [101,102]. Moreover, one recent RCT on bipolar patients showed that add-on *Lactobacillus* probiotics after hospital discharge was associated with lower rehospitalization rates compared to placebo [103]. Interestingly, this study also found that the putative preventive effect was greater in those with elevated levels of inflammation pre-treatment, again suggesting that there may be a subgroup of mood disorder patients with inflammation that would benefit more from this type of intervention [103]. A recent small-scale pilot RCT tested the efficacy of *Lactobacillus Reuteri (L. Reuteri)* versus placebo in war veterans with Post-traumatic Stress Disorder symptoms [104]. Although this study was not powered to detect treatment-related changes in clinical PTSD symptoms, assignment to the *L. Reuteri* arm was associated with a decrease in plasma CRP levels that approached statistical significance, suggesting that CRP may be a useful biomarker to monitor treatment effect in future studies.

In the first probiotics RCT that included patients with DSM-IV diagnosed depression of at least moderate severity, Akkasheh et al. [105] randomized 40 unmedicated MDD patients to receive either a combined probiotic supplement containing each of the following strains: *Lactobacillus acidophilus*, *Lactobacillus casei*, and *Bifidobacterium bifidum*, or placebo. After eight weeks of treatment, depressive symptoms had significantly decreased in the treatment compared to the placebo group. Additionally, in the intervention group, there was a statistically significant decrease in hs-CRP and insulin levels. Thus, this study suggests that some of the antidepressant effects of probiotics may be mediated via anti-inflammatory mechanisms and beneficial effects on metabolic markers. Since three different strains of bacteria were used in combination it is, however, not possible to ascertain whether one of these strains conveyed the beneficial effects, and if so which one. While this study had several strengths, it also came with several notable caveats; for instance, the authors recognized the need for larger sample sizes in future studies, longer follow-up time, and a possibility to analyze more biomarkers of inflammation and oxidative stress. In another randomized placebo-controlled trial including 40 unmedicated patients with irritable bowel syndrome (IBS) and comorbid MDD, 90 days of treatment with *Bacillus coagulans (MTCC 5856)* resulted in a significant reduction of both depressive and IBS symptoms compared to placebo [106]. This study also found a significant reduction in serum myeloperoxidase, a marker of oxidative stress and inflammation [107,108] over the course of treatment, and this change was not seen in the placebo group. These findings are mainly in line with another RCT on IBS and comorbid MDD, reporting a decrease in depressive symptoms, as well as improvement in quality of life, in patients treated with *Bifidobacterium longum* compared to placebo [109]. No effects on anxiety or IBS symptoms were, however, observed in this study, suggesting that any effect is restricted to depressive symptoms and quality of life. Interestingly, patients assigned to probiotics had reduced limbic activity (reduced response to fearful stimuli), per fMRI analyses, further strengthening the hypothesis that the gut-brain axis may be a viable target in the treatment of psychiatric disorders. In this study, there were no significant between-group differences in inflammatory markers or microbiota profiles. Another RCT [110] found that 8 weeks of add-on treatment with probiotics (*Lactobacillus helveticus* and *Bifidobacterium longum*) resulted in a significant decrease of depressive symptoms compared to both placebo and prebiotics. Moreover, those who were randomized to the probiotics group showed significantly decreased serum kynurenine/tryptophan ratio compared to the placebo group at the end of the study, when adjusted for serum isoleucine. The rationale for controlling for isoleucine was that branched-chain amino acids like isoleucine and tryptophan compete for the same transporter from blood to brain. The authors suggested that a decrease in the kynurenine/tryptophan ratio might indicate an effect of probiotics on the metabolism of tryptophan, potentially by reducing the activity of enzymes responsible for conversion of tryptophan to kynurenine and that this might partly explain the observed effect on depressive symptoms.

Based on the literature review above, it seems that future probiotics studies should include patients with clinical, DSM-verified MDD. Moreover, future studies should investigate mechanisms mediating any antidepressant effect of probiotics, including the use of brain imaging as well as relevant blood and feces biomarkers. Finally, the heterogeneity of symptoms and mechanisms in MDD has hampered the development of targeted treatments. One way to overcome this is to use “enriched population study design”. This strategy decreases heterogeneity and can reduce costs, improve risk-benefit relationships, and has been advocated as a means to advance “personalized psychiatry”, particularly in the context of inflammatory depression [8]. Based on the biological mechanisms of probiotics described above, future studies should consider enriching patients based on inflammatory markers in order to identify individuals who are more likely to respond to this particular intervention.

### 3.3. Is Exercise Efficacious in Inflammatory Depression?

As an alternative to more conventional antidepressants, exercise has gained increasing attention as a potential preventive measure, or even a treatment option, for some cases of MDD. Several meta-analyses have provided evidence that physical activity has an antidepressant effect [111,112,113], and it is now recommended in the National Institute for Health and Care Excellence (NICE) guidelines for mild to moderate depression [114]. Some critical reviews of the literature have, however, noted that many of the exercise treatment trials have a high risk of bias, and when these are eliminated from the analyses, the antidepressant effect is reduced [115]. Again, this view may be consistent with the heterogeneous nature of depression, and that a certain intervention may not benefit all depressed patients, but only a subset. Interestingly, recent preclinical and clinical research suggests that the antidepressant effect of physical activity may be mediated via anti-inflammatory mechanisms [116]. For instance, in a chronic unpredictable mild stress (CUMS) paradigm, exercise alleviated depressive-like behavior in mice, and also downregulated peripheral and central inflammatory markers associated with CUMS [117]. Interestingly, exercise also improved intestinal function and increased central dopamine expression in these animals, suggesting that exercise may, in addition to having an anti-inflammatory effect, also exert its antidepressant effects via mechanisms involving gut permeability and dopamine metabolism. As described below, preliminary clinical data also suggest that exercise, to the extent that it has an antidepressant effect, is more efficacious in depressed patients with signs of low-grade inflammation.

Rethorst et al. (2013) investigated the association between proinflammatory cytokines (IFN-γ, TNF-α, IL-6, and IL-1β) and antidepressant response to exercise in MDD patients who were all partial SSRI responders [118]. A total of 126 MDD patients were randomized to receive either low or high dose of exercise (4 or 12 kilocalories per kilogram of body weight per week respectively) for 12 weeks, and raters of depressive symptoms were blinded to group assignment. They found that high levels of TNF-α at baseline predicted a greater decrease in depressive symptoms after 12 weeks of physical activity, suggesting a larger antidepressant effect in MDD patients with elevated inflammation at baseline. There was also a positive correlation between a decrease in IL-1β levels over the course of treatment and a decrease in depressive symptoms. The association between change in IL-1β levels and depressive symptoms remained significant in the high-dose but not in the low-dose exercise group, suggesting a dose-response relationship. However, physical activity did not significantly decrease any of cytokines over the course of the study, regardless of exercise dose. In another study by Lavebratt et al. (2017), patients with mild to moderate depression were randomized to light, moderate, or vigorous exercise (three 60-min classes a week) [119]. They found that higher baseline levels of IL-6 were related to greater improvement in depressive symptoms after 12 weeks of physical activity. Again, this strengthens the hypothesis of a more favorable effect of exercise in “inflammatory depression” compared to non-inflammatory depression. This study also found that IL-6 levels decreased significantly from baseline to post-treatment when all participants were analyzed together regardless of exercise intensity, and there was also a positive correlation between change in depression symptoms and change in IL-6 level over the course of treatment. However, no significant effect of different exercise intensities on change in IL-6 levels could be found, possible due to limited sample sizes.

The mechanisms underlying an antidepressant effect of exercise remains understudied. However, physical activity has been associated with various positive immune-modulating properties like leukocyte redistribution and trafficking, maturation, and functions of different immune cells [120]. The putative anti-inflammatory properties of exercise can partially be explained by reduced visceral adiposity, as adipose tissue is an important source of systemic inflammation [121]. Some studies have suggested that physical activity increases gene expression of PGC-1-α (peroxisome proliferator-activated receptor C coactivator-1-α) [122], a transcription coactivator that plays a central role in the regulation of cellular energy metabolism [123]. PGC-1-α is a coactivator of transcription factors with neuroprotective properties and controls gene expression of pro-inflammatory cytokines in muscles, partly by inhibition of the NF-KB pathway [116,124] (11,8). Recent research has also suggested that PGC-1-α facilitates the transformation of peripheral kynurenine to kyn-a [122]. Unlike kynurenine, kyn-a does not pass the BBB and consequently prevents the potentially neuroinflammatory effects of central tryptophan metabolism along the kynurenine pathway [122].

To summarize, preliminary evidence suggests that exercise may play a role in the treatment of inflammatory depression, although future large-scale and well-designed clinical trials are needed. Selected key studies on treatment of inflammatory depression reviewed in this paper are summarized in Table 1.

## 4. Conclusions

We here posit that inflammatory depression is a specific subtype that responds to specific mechanism-based treatments. We have reviewed several upstream, potentially causal, mechanisms driving low-grade inflammation in this subtype of depression. We have also discussed downstream mechanisms mediating the link between inflammation and symptoms of depression, including alterations in dopaminergic neurotransmission and tryptophan metabolism. There is preliminary evidence that inflammatory depression can be addressed by specific treatment choices targeting some of these pathways, including the gut microbiota and omega-3 fatty acids. In future studies, it will be important to determine how the effect of such non-pharmacological interventions compare to potentially more potent pharmacological interventions. For instance, both anti-inflammatory drugs [11] and certain conventional antidepressants [126,127] may be efficacious in inflammatory depression. Furthermore, forthcoming studies might consider stratifying subjects based on severity or duration of illness, since such factors may contribute to inflammatory depression [128]. Finally, future clinical trials should enrich patient populations for inflammatory markers or use “match/mismatch designs” in order to definitely test the hypothesis that inflammatory depression is a clinically meaningful construct with the potential to advance precision psychiatry.

## Figures and Tables

**Figure 1 ijms-22-01640-f001:**
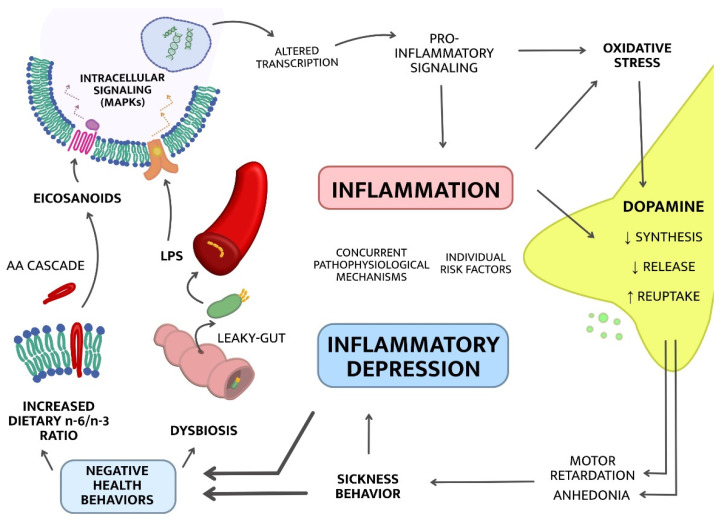
Hypothetical model of bidirectional interactions between systemic low-grade inflammation and depression. From bottom left: Negative health behaviors such as poor dietary habits, stress, and a sedentary lifestyle could trigger and/or enhance inflammatory responses. Below, we discuss two potential mechanisms mediating the effects of such behaviors on inflammation; increased dietary n-6/n-3 ratio and dysbiosis. Subsequent pro-inflammatory signaling may trigger oxidative stress and lead to alterations in dopamine release, reuptake and synthesis. This in turn may generate symptoms of psychomotor retardation and motivational anhedonia, further reinforcing a state of sickness behavior, and may induce, enhance, or maintain depression and more negative health behaviors. Abbreviations: omega-6 (n-6), omega-3 (n-3), arachidonic acid (AA), lipopolysaccharide (LPS), mitogen-activated protein kinases (MAPK).

**Figure 2 ijms-22-01640-f002:**
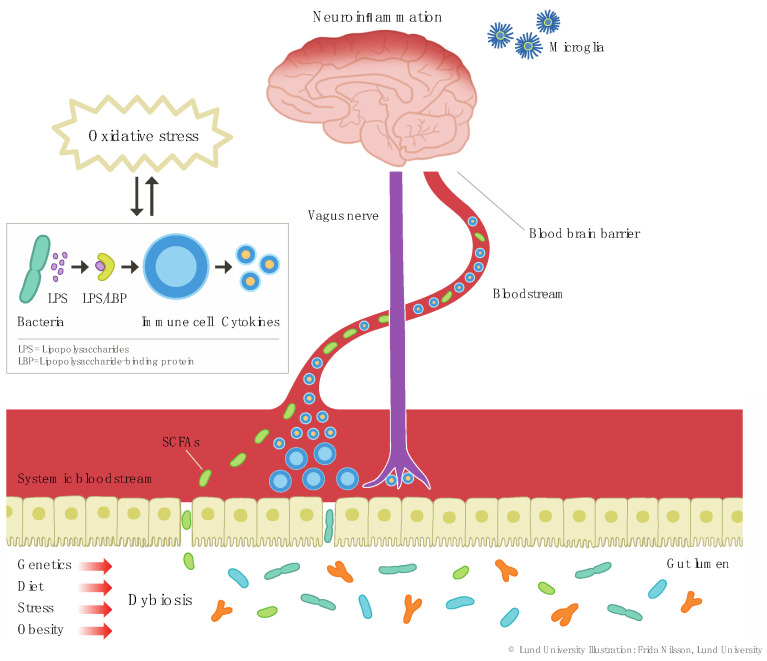
Dysbiosis and increased gut permeability may lead to translocation of LPS from the gut lumen into the circulation triggering low-grade inflammation. SCFAs also translocate into circulation and may have several beneficial effects including enhancing the integrity of the intestinal barrier and the BBB. Circulating pro-inflammatory cytokines either signal to the brain via the vagal nerve or cross the BBB and thereby contribute to neuroinflammation and microglia activation. Abbreviations: lipopolysaccharides (LPS), short-chain fatty acids (SCFAs), lipopolysaccharide-binding protein (LBP), blood-brain barrier (BBB).

**Figure 3 ijms-22-01640-f003:**
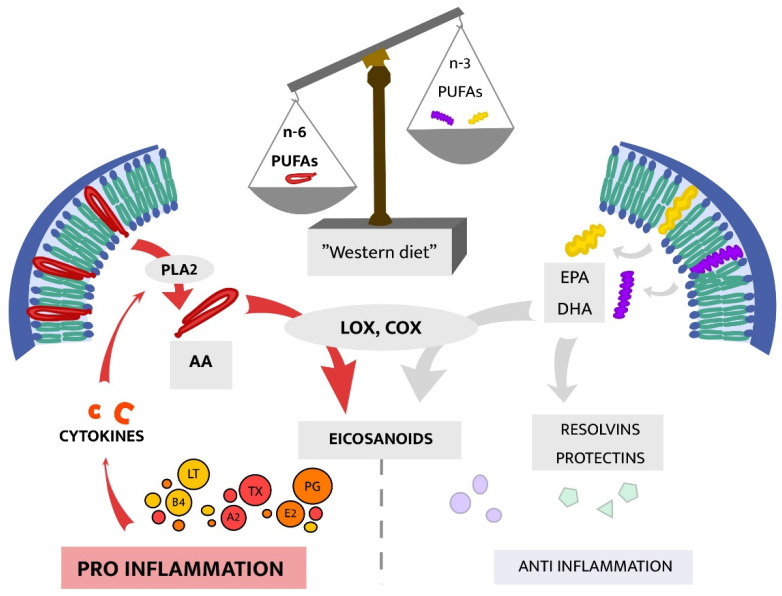
Enzymes LOX and COX are involved in various steps of PUFA metabolization, affecting the pro/anti-inflammatory balance. Abbreviations: omega-6 (n-6), omega-3 (n-3), polyunsaturated fatty acids (PUFAs), eicosapentaenoic acid (EPA), docosahexaenoic acid (DHA), phospholipidase 2 (PLA2), arachidonic acid (AA), lipoxygenase (LOX), cyclooxygenase (COX), prostaglandines (PG), thromboxanes (TX), leukotrienes (LT).

**Figure 4 ijms-22-01640-f004:**
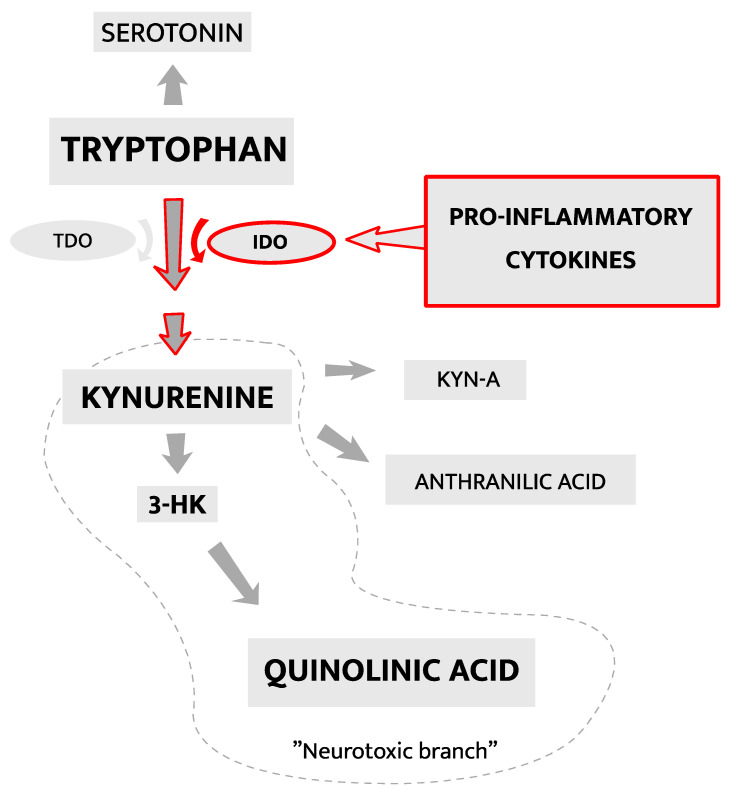
Metabolization of the essential amino acid tryptophan is shown in a simplified schematic with focus on the metabolites discussed in this review. Marked in red is the activation of IDO by pro-inflammatory cytokines, triggering the formation of kynurenine from tryptophan. This is the first and rate-limiting step of the kynurenine pathway. Abbreviations: tryptophan 2,3-dioxygenase (TDO), indoleamine 2,3-dioxygenase (IDO), 3-hydroxykynurenine (3-HK), kynurenic Acid (Kyn-a).

**Figure 5 ijms-22-01640-f005:**
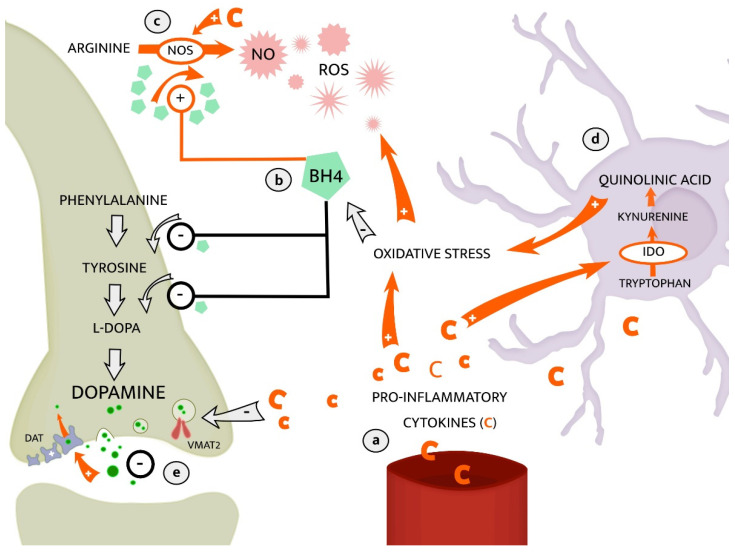
Crosstalk between the immune and dopaminergic systems. A pro-inflammatory state is triggered by cytokines produced by microglia or entering the central nervous system (CNS) from the peripheral circulation (**a**). Cytokine-actions with increased oxidative stress through formation of ROS may reduce part of the BH4 pool, resulting in less availability of BH4 for dopamine synthesis (**b**). BH4 availability is further reduced when cytokines activate NOS, demanding BH4 as co-factor in production of NO, further adding to oxidative stress (**c**). To the right in the figure, the “neurotoxic” and cytokine-induced branch of kynurenine pathway is presented, which occurs in microglia (**d**). The KP-metabolite quinolinic acid has neurotoxic properties and adds to oxidative stress. Lastly, available dopamine in pre-synaptic clefts is further reduced as cytokines both decrease VMAT2’s packaging of dopamine into vesicles for release, and increase DAT expression and function for re-uptake (**e**) [67]. Abbreviations: nitric oxide synthase (NOS), nitric oxide (NO), reactive oxygen species (ROS), tetrahydrobiopterin (BH4), indoleamine 2,3-dioxygenase (IDO), vesicular monoamine transporter 2 (VMAT2), dopamine transporter (DAT).

**Table 1 ijms-22-01640-t001:** Characteristics of key studies on treatment of inflammatory depression.

Reference	Sample	Study Type	Biomarkers	Main Findings	Notes and Limitations
Rapaport et al., 2016 [85]	N = 155, all MDD and unmedicated	RCT, 8 with tx: EPA, DHA, or placebo	IL-1, IL-6, CRP, leptin, adiponectin	No overall tx effects. Subjects with high inflammation improved more on EPA than placebo	Evidence for dose-response effect with increasing EPA-placebo separation with increasing number of elevated inflammation markers. Proof-of-concept study with post hoc analyses in need of replication.
Akkasheh et al., 2016 [105]	N = 40, all MDD and unmedicated	RCT, 8 with tx: probiotics or placebo	Hs-CRP, p-glucose, total antioxidant capacity, lipids and more	Probiotic tx associated with significant improvement in depressive symptoms and decreased insulin and CRP levels	Probiotic strain not specified. Small sample size, in need of replication to study effect on inflammatory markers and lipids
Rudzki et al., 2018 [125]	N = 79, all MDD and SSRI-tx	RCT, 8 with tx: probiotics or placebo	IL-6, IL-1β, TNF-α, CRP, cortisol, TSH, kynurenine pathway-metabolites	Probiotic tx not associated with significant antidepressant or anti-inflammatory effect. Decrease in kynurenine concentration and improvement of cognitive function in the probiotic group	To our knowledge, 1^st^ evidence for the effect of a probiotic intervention on the kynurenine pathway.
Rethorst et al., 2013 [118]	N = 105, all MDD and SSRI-tx.	Subjects randomly assigned to 12-week: low or high dose exercise intervention. Raters blinded to group assignment	IFN-γ, IL-6, IL-1β, TNF-α	TNF-α levels at baseline predicted better outcome. Delta depressive symptoms significantly correlated with delta IL-1β	1^st^ study to show that exercise may have greater antidepressant efficacy in inflammatory depression. Exercise did not lower cytokine levels. Lacked control group.
Lavebratt et al., 2017 [119]	N = 116, all MDD (follow-up data, n = 89)	Secondary analysis of RCT comparing exercise and ICBT. 12 w exercise intervention: light, moderate or vigorous	IL-6	Higher baseline levels of IL-6 associated with greater improvement in depressive symptom severity. Positive correlation (*p* = 0.049) between reduced symptoms and reduction in IL-6 level	No significant effect of exercise intensities on IL-6 change. Exercise outside the supervised exercise not monitored.

Abbreviations: major depressive disorder (MDD), eicosapentaenoic acid (EPA), docosahexaenoic acid (DHA), high-sensitivity CRP (Hs-CRP), Internet-based cognitive behavioral therapy (ICBT).

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
