# Peer review of "Inflammatory Depression—Mechanisms and Non-Pharmacological Interventions"

_ijms, 2021, doi:10.3390/ijms22041640_

Round 1

Reviewer 1 Report

  • This is a non-systematic review. As such, it is difficult for the final reader to rank the quality of the appraised evidence. On the contrary, a non-boundary review can include multiple records in the same summary, which is a plus for topics with a bounce of evidence.
  • The results largely rely on pre-clinical models of inflammatory depression, which is fine. However, the sections about the potential therapeutic avenues (e.g., paragraph n.3.2 onward), properly rely on clinical trials. Even if they are proof of concept or preliminary studies. My quibble is that I would expect the results to provide clear-cut discrimination between animal and human studies, or to rather comment on what is known about whom.
  • The conclusions are too concise: please consider comparing the potential non-pharmacological therapeutic options against the pharmacological ones already appraised in the existing literature (there is plenty of evidence on the matter). Please consider ranking the evidence about the quality, for example by discriminating RCT vs. non-RCT trials, and to account for the different stages of MDD, which may also reflect different inflammatory status.
  • I understand that the in-house editorial guidelines do not require a methods section before the results one. However, even a brief section appended after the conclusion should be needed, no matter if the present report is a narrative one. This would be appreciated by the final reader concerned about selective reporting bias.

Author Response

Dear dr Stankovic,

We hereby submit a revised version of our manuscript entitled “Inflammatory depression - mechanisms and non-pharmacological interventions” authored by Klara Suneson, Jesper Lindahl, Simon Chamli Hårsmar, Gustav Söderberg and Daniel Lindqvist, for your consideration for publication in IJMS.

We would like to thank the reviewers for their valuable comments that have significantly improved the quality of the manuscript. Our detailed responses to the reviewers’ critiques are given in italics in the response letter attached below. We have revised the manuscript using track changes to highlight any changes. In addition to responding to the issues raised by reviewers, we have also made a minor modification to Figure 2, and have therefore uploaded a new version of that figure in the submission system (attached to response to reviewer 1).

 We now hope that the paper is suitable for publication in IJMS.  

Yours sincerely,

Klara Suneson and Daniel Lindqvist

Comments from reviewers:

-Reviewer 1

This is a non-systematic review. As such, it is difficult for the final reader to rank the quality of the appraised evidence. On the contrary, a non-boundary review can include multiple records in the same summary, which is a plus for topics with a bounce of evidence.

The results largely rely on pre-clinical models of inflammatory depression, which is fine. However, the sections about the potential therapeutic avenues (e.g., paragraph n.3.2 onward), properly rely on clinical trials. Even if they are proof of concept or preliminary studies. My quibble is that I would expect the results to provide clear-cut discrimination between animal and human studies, or to rather comment on what is known about whom.

Authors response: Thanks very much for raising this issue. As the reviewer points out, in the mechanistic part of the paper we mainly discuss experimental data, while the sections on therapeutics mostly deals with clinical data. In section 3.2 we review clinical probiotics studies, although the section starts with a brief overview of some pertinent animal studies, and we also give a few examples of preclinical studies later on when we discuss the clinical effects of probiotics (lines 469-474). We believe that this strategy increases the readability and properly sets the state for the clinical part of the section. However, we agree that this distinction could be made clearer, and we have therefore separated these two subsections (line 439) and also added several sentences for clarification (see lines 441-442, 469).

The conclusions are too concise: please consider comparing the potential non-pharmacological therapeutic options against the pharmacological ones already appraised in the existing literature (there is plenty of evidence on the matter). Please consider ranking the evidence about the quality, for example by discriminating RCT vs. non-RCT trials, and to account for the different stages of MDD, which may also reflect different inflammatory status.

Authors response: Thanks for these comments. We have now modified the conclusion in accordance with these comments (see lines 614-620). We have also updated some of the text and Table 1 to clarify which studies are RCT and which are not (see table 1 on page 15-16 and revisions in the text; line 564-565).

I understand that the in-house editorial guidelines do not require a methods section before the results one. However, even a brief section appended after the conclusion should be needed, no matter if the present report is a narrative one. This would be appreciated by the final reader concerned about selective reporting bias.

Authors response: We have now added a sentence about this on page 17, lines 624-625.

Reviewer 2 Report

The review by Suneson et al. describes in a very exhaustive way the correlation between systemic inflammatory states and the appearance of depressive disorders both in mouse models and in humans, a process that the authors define as Inflammatory Depression. Furthermore, the authors analyze in detail the molecular pathways underlying the inflammatory hypothesis of depression. In the second part of the review, the authors focus on the role of three different non-pharmacological treatments (omega-3, probiotics and physical activity) capable of alleviating inflammatory symptoms and consequently the  related  depressive disorders. The review is written in a very clear and explanatory way, it highlights the most significant results and the open question on this topic and is correlated by a rich and appropriate scientific literature. 

Just a couple of considerations:

In the section relating to Therapeutic Implications, the authors provide an excursus of the main results on the role of omega-3, probiotics and physical activity in countering inflammatory phenomena and depressive disorders. In the first two cases (omega-3 and probiotics) the Authors describe studies on mouse models and on humans, while for physical activity role only works on humans are reported. It would be interesting to integrate this part also with studies on mouse models to more homogenize the different sections of the review.

Furthermore, are there studies (on mouse models or on humans) in which a combination of the various treatments, omega-3 and physical activity or probiotics and physical activity, have been used to evaluate their potential additive synergistic effect?

Author Response

Reviewer 2

The review by Suneson et al. describes in a very exhaustive way the correlation between systemic inflammatory states and the appearance of depressive disorders both in mouse models and in humans, a process that the authors define as Inflammatory Depression. Furthermore, the authors analyze in detail the molecular pathways underlying the inflammatory hypothesis of depression. In the second part of the review, the authors focus on the role of three different non-pharmacological treatments (omega-3, probiotics and physical activity) capable of alleviating inflammatory symptoms and consequently the  related  depressive disorders. The review is written in a very clear and explanatory way, it highlights the most significant results and the open question on this topic and is correlated by a rich and appropriate scientific literature. 

Just a couple of considerations:

In the section relating to Therapeutic Implications, the authors provide an excursus of the main results on the role of omega-3, probiotics and physical activity in countering inflammatory phenomena and depressive disorders. In the first two cases (omega-3 and probiotics) the Authors describe studies on mouse models and on humans, while for physical activity role only works on humans are reported. It would be interesting to integrate this part also with studies on mouse models to more homogenize the different sections of the review.

Authors response: Thank you for this very relevant suggestion. We have now added the following sentence to page 14:

“For instance, in a chronic unpredictable mild stress (CUMS) paradigm, exercise alleviated depressive-like behavior in mice, and also downregulated peripheral and central inflammatory markers associated with CUMS (Liu et al., 2020) Interestingly, exercise also improved intestinal function and increased central dopamine expression in these animals, suggesting that exercise might exert its antidepressant effects via mechanisms involving inflammation, gut permeability and dopamine metabolism.”

Furthermore, are there studies (on mouse models or on humans) in which a combination of the various treatments, omega-3 and physical activity or probiotics and physical activity, have been used to evaluate their potential additive synergistic effect?

Authors response: This is a very interesting research question that has not, to the best of our knowledge, been tested so far in depression.